# Theoretical and Experimental Studies of CO_2_ Absorption in Double-Unit Flat-Plate Membrane Contactors

**DOI:** 10.3390/membranes12040370

**Published:** 2022-03-29

**Authors:** Chii-Dong Ho, Hsuan Chang, Jr-Wei Tu, Jun-Wei Lim, Chung-Pao Chiou, Yu-Jie Chen

**Affiliations:** 1Department of Chemical and Materials Engineering, Tamkang University, Tamsui, New Taipei 251, Taiwan; nhchang@mail.tku.edu.tw (H.C.); 891360033@gms.tku.edu.tw (J.-W.T.); a0937420623@gmail.com (C.-P.C.); mini1752@gmail.com (Y.-J.C.); 2Department of Fundamental and Applied Sciences, HICoE-Centre for Biofuel and Biochemical Research, Institute of Self-Sustainable Building, Universiti Teknologi PETRONAS, Seri Iskandar 32610, Perak Darul Ridzuan, Malaysia; junwei.lim@utp.edu.my

**Keywords:** turbulence promoter, carbon dioxide absorption, Sherwood number, double-unit membrane contactor, concentration polarization

## Abstract

Theoretical predictions of carbon dioxide absorption flux were analyzed by developing one-dimensional mathematical modeling using the chemical absorption theory based on mass-transfer resistances in series. The CO_2_ absorption into monoethanolamine (MEA) solutions was treated as chemical absorption, accompanied by a large equilibrium constant. The experimental work of the CO_2_ absorption flux using MEA solution was conducted in double-unit flat-plate membrane contactors with embedded 3D turbulence promoters under various absorbent flow rates, CO_2_ feed flow rates, and inlet CO_2_ concentrations in the gas feed stream for both concurrent and countercurrent flow operations. A more compact double-unit module with embedded 3D turbulence promoters could increase the membrane stability to prevent flow-induced vibration and enhance the CO_2_ absorption rate by overwhelming the concentration polarization on the membrane surfaces. The measured absorption fluxes with a near pseudo-first-order reaction were in good agreement with the theoretical predictions for the CO_2_ absorption efficiency in aqueous MEA solutions, which was shown to be substantially larger than the physical absorption in water. By embedding 3D turbulence promoters in the MEA feed channel, the new design accomplishes a considerable CO_2_ absorption flux compared with an empty channel as well as the single unit module. This demonstrates the value and originality of the present study regarding the technical feasibility. The absorption flux enhancement for the double-unit module with embedded 3D turbulence promoters could provide a maximum relative increase of up to 40% due to the diminution in the concentration polarization effect. The correlated equation of the average Sherwood number was obtained numerically using the fourth Runge–Kutta method in a generalized and simplified expression to calculate the mass transfer coefficient of the CO_2_ absorption in the double-unit flat-plate membrane contactor with turbulence promoter channels.

## 1. Introduction

CO_2_ capture techniques [1] to remove CO_2_ from gas mixtures for industrial processes have been widely studied over the past decade, including conventional contactors with chemical absorbents. Membrane contactors involve the combined techniques of conventional gas absorption and membrane separation, and they can effectively overcome the problems associated with conventional contactors because of the presence of a membrane [2]. Several technologies, namely absorption [3], adsorption [4], and membrane processes [5], are the processes capable of CO_2_ absorption, for which the membrane contactor is a promising technology with high absorption efficiency. The membrane contactor offers the advantages of low energy consumption, a large and stable gas-liquid contact area, continuous operations, high modularity, and flexibility scale-up [6]. The configuration of the membrane contactor in which the hydrophobic porous membrane is employed acts as a barrier separating CO_2_ gas feed and absorbent streams, and the gas/liquid interface is formed at the membrane pore mouth in the gas feed stream because of the non-wetted hydrophobic porous membrane. Currently, chemical absorption by amine solution is the most advanced technology for absorbing CO_2_ from gas mixtures, as confirmed by a previous study [7], and alkanolamine-based CO_2_ absorption processes have been used commercially. Faiz and Al-Marzouqi [8] developed a mathematical model for CO_2_ absorption using monoethanolamine (MEA) as an absorbent from natural gas at high pressures, and successful process intensifications for CO_2_ absorption processes have been investigated that employ selective membrane materials [9]. Moreover, the membrane absorption efficiency dependent on the distribution coefficient was determined with lower membrane wettability [10] and the properties of absorbents [11]. CO_2_ is transported from the gas side across the boundary layer and membrane to the absorbent side, and thus both chemical absorption and separation occur simultaneously according to the diffusion–reaction model [12,13]. Theoretical studies of the dusty gas model [14], including Knudsen–molecular diffusion, were developed to comprehensively understand the mass transfer behavior of CO_2_ absorption [15] using amines [16] and analyzing the CO_2_ absorption efficiency.

The concentration polarization effect [17] builds concentration gradients in the turbulent boundary layer region adjacent to the membrane surface, which can cause a considerable reduction in the mass-transfer rate [18] and thus decrease the separation efficiency [19] of most membrane separation processes, such as gas absorption [20], reverse osmosis [21], extraction [22], and dialysis [23]. A depletion of CO_2_ and an accumulation of the permeating CO_2_ occurred concurrently in the mass-transfer boundary layers adjacent to both membrane surfaces. The accumulation of CO_2_ at the membrane surface contributed to this, resulting in the reduction of the imposed driving force and, consequently, the absorption flux. An effective strategy was investigated to capture CO_2_ in turbulent flow conditions [24] by embedding spiral wires into the flow channel [25] compared with considering a laminar flow velocity of the liquid profile [26]. The absorption efficiency in a parallel-plate gas/liquid polytetrafluoroethylene (PTFE) membrane contactor was augmented by inserting turbulence promoters [27]. The present work studied the overall mass transfer resistance reduction and aimed to boost the turbulent intensity by embedding 3D turbulence promoters, which resulted in a lessening of the mass transfer boundary layer to accelerate CO_2_ transport in absorbent solutions. This implementation reduces the overall separation efficiency and permeate flux [19] owing to the decrease in the available concentration driving force of the permeating CO_2_ across the membrane in the mass transfer boundary layer. Various actions to minimize the concentration polarization to augment a larger turbulence intensity were taken by implementing eddy promoters into the flow channels, with numerous advantages [28], such as spacer filaments [29] and carbon-fiber spacers [30], and thus a higher convective mass transfer coefficient was effectively achieved by enhancing the turbulent intensity [31].

This study discusses the mathematical modeling of CO_2_ absorption in MEA solution as an absorbent and the device performance improvement achieved by embedding 3D turbulence promoters in double-unit flat-plate membrane contactors under both concurrent flow and countercurrent flow operations. The one-dimensional steady-state modeling equation was successfully applied to predict the CO_2_ absorption flux under various operational conditions associated with the reactions that occurred [15] using amines as absorbents [32]. The mechanisms of CO_2_ absorption in MEA solution were investigated in previous research [33]. A theoretical model was developed to analyze the decrease in the concentration polarization effect for gas/liquid membrane contactors with embedded 3D turbulence promoters by creating eddies in the feed stream, which disrupt the boundary layer and yield a higher CO_2_ absorption rate. The magnitude of the concentration polarization coefficient $\gamma_{m}$ plays an important role in examining the device performance in the CO_2_/MEA membrane absorption module. A higher value of $\gamma_{m}$ denotes a larger mass flux of CO_2_ transferring from the gas side to the MEA feed stream, which was accomplished by diminishing the undesirable influence on the mass transfer rate due to the disruption of the concentration boundary layer by embedding 3D turbulence promoters. An attempt to augment the turbulence intensity by embedding 3D turbulence promoters into the flow channel was implemented in the present study, and thus a higher device performance was achieved, which was economical in terms of absorption efficiency. The concentration polarization coefficient was calculated theoretically, and its relationship with other parameters was verified using the theory developed in this study. The turbulence intensity augmented by embedding 3D turbulence promoters in the MEA feed stream was examined with the experimental results. Thus, a modified Sherwood number, adapted in terms of a key dimensionless quantity and named the mass-transfer enhancement factor, was incorporated and regressed a correlated expression of the convective mass transfer coefficient using various parameters, such as the geometric shapes of the turbulence promoters, flow patterns, flow types, inlet concentrations, and MEA feed flow rates. The suitable selection of CO_2_ absorption flux improvement and increased power consumption by considering the economic assessment of the module designs and system operations is also discussed.

## 2. Experimental Apparatus

The fabrication scheme and the schematic detailed configuration of a double-unit flat-plate membrane contactor module for CO_2_ absorption by MEA absorbent are illustrated in Figure 1 and Figure 2, respectively.

A photo of the operating experimental apparatus is shown in Figure 3, with acrylic plates used as outside walls screwed tightly by two parallel stainless-steel plates. The double-unit flat-plate membrane contactor module contains two flow channels with 3D turbulence promoters embedded onto the MEA feed stream and another empty channel for the CO_2_/N_2_ gas mixture. Two parallel-plate flow channels (*L* = 0.21 m, *W* = 0.29 m, *H* = 1 mm) separated by a hydrophobic composite membrane made of PTFE/PP (polytetrafluoroethylene/polypropylene) as the permeating medium with a nominal pore size of 0.1, a porosity of 0.72, and a thickness of 130 µm (ADVANTEC) were used in the experimental work. The membrane surface in the empty channel was wound with 0.2 mm nylon fiber as a supporting material to prevent vibration.

Two geometric shapes (Circle and Square) were embedded into the MEA feed flow channel with two array patterns for comparisons of the CO_2_ absorption rate using an amine solution in the device performance, as shown in Figure 4. The aqueous MEA solution passing through the channel with embedded 3D turbulence promoters was regulated by a flow meter (MB15GH-4-1, Fong-Jei, New Taipei, Taiwan) as the liquid absorbent was pumped from a reservoir. The experimental runs were carried out with 30 wt% MEA ($5.0\times 10^{3}$ mol/m^3^) for various MEA feed flow rates within the range of 5–10 cm^3^/s (5.0, 6.67, 8.33, and 10.0 cm^3^/s). A gas mixture containing CO_2_/N_2_ introduced from the gas mixing tank (EW-06065-02, Cole Parmer Company, Vernon Hills, IL, USA) was regulated using the mass flow controller (N12031501PC-540, Protec, Brooks Instrument, Hatfield, PA, USA) at 5 cm^3^/s with three inlet CO_2_ concentrations of 30%, 35%, and 40%. The outlet CO_2_ concentrations were measured using gas chromatography (Model HY 3000 Chromatograph, China Corporation) and recorded to calculate the CO_2_ absorption flux for comparisons under various operating conditions.

## 3. Mathematical Modeling

A mass transfer behavior analysis was conducted to describe the concentration gradient between the membrane surfaces due to the mass flux transfer from the CO_2_ gas side to the MEA feed side. The balances of mass flux due to mass diffusion and chemical reaction by the mass conservation were formulated simultaneously. The steady-state isothermal diffusion–reaction process in the membrane contactor module causes the trans-membrane mass flux of CO_2_ dominated by the concentration boundary layers on both bulk streams, the properties of the membrane, and the operating conditions. The CO_2_ concentration on the membrane–liquid interface was determined by the dimensionless Henry’s law constant: $H_{C}=C_{2}/C_{1}=0.73$ [33].

The mass balances of the gas/liquid membrane contactor were described on the basis of the principle of isothermal processes, performing the mass balance of mass flux conservation in each mass transfer region: (a) the CO_2_ feed stream; (b) the hydrophobic composite membrane, and (c) the MEA feed stream. The mass flux balance equations were derived for each mass transfer region under steady-state operation according to the schematic diagram of the single unit (approximately one half) of the double-unit gas/liquid membrane contactor module in Figure 5a, and was depicted in Equations (1)–(3) by the concentration driving force gradient as follows:(1)$$J_{g}=k_{a}\left(C_{a}-C_{1}\right)$$
(2)$$J_{\ell }=k_{b}\left(\frac{{K^{\prime }}_{ex}C_{2\left(\ell \right)}}{H_{c}}-\frac{C_{b\left(\ell \right)}}{H_{c}}\right)$$

The mass transfer behavior in the membrane was investigated [34] according to dusty gas model [8], and the mass flux of CO_2_ diffusing through the trans-membrane was evaluated using the membrane permeation coefficient ($c_{m}$) and the saturation partial pressure differences (ΔP) [35] as:(3)$$J_{m}=c_{m}(P_{1}-P_{2})\frac{1}{M_{w}}={\left.c_{m}\frac{dP}{dC}\;\right|}_{C_{mean}}(C_{1}-C_{2(g)})\frac{1}{M_{w}}=c_{m}RT(C_{1}-\frac{{K^{\prime }}_{ex}C_{2\left(\ell \right)}}{H_{c}})\frac{1}{M_{w}}=K_{m}(C_{1}-\frac{{K^{\prime }}_{ex}C_{2\left(\ell \right)}}{H_{c}})$$
in which $K_{m}$ is the overall mass transfer coefficient of the membrane, and the reduced equilibrium constant at *T* = 298 K [36] and the membrane permeation coefficient [37] with the tortuosity τ=1/ε [38] were determined as:(4)$$K_{ex}^{\prime }=K_{ex}{[\mathrm{MEA}]/[H}^{+}],K_{ex}={[\mathrm{MEACOO}}^{-}{]\;[H}^{+}{]/[\mathrm{CO}}_{2}][\mathrm{MEA}]=1.25\times 10^{-5}$$
(5)$$c_{m}={\left(\frac{1}{c_{K}}+\frac{1}{c_{M}}\right)}^{-1}={\left\{{\left[1.064\frac{\varepsilon \;r_{p}}{\tau \delta_{m}}{\left(\frac{M_{w}}{RT_{m}}\right)}^{1/2}\right]}^{-1}+{\left[\frac{1}{{\left|Y_{m}\right|}_{\ln}}\frac{D_{m}\varepsilon }{\delta_{m}\tau }\frac{M_{w}}{RT_{m}}\right]}^{-1}\right\}}^{-1}$$

Equating the amount of mass flux was performed by the conservation law in three regions transferred through the gas feed side, the membrane pores, and the liquid feed side for the modules with or without embedded 3D turbulence promoters, as shown in Figure 5b.
(6)$$J_{i}=J_{g}=J_{m}=J_{\ell }\;i=promoter,empty\;$$

The modeling equations of mass balances of the gas feed and liquid feed streams were derived by making the mass flux diagram presented in a finite control element under concurrent flow and countercurrent flow operations in Figure 6a,b, respectively, giving:(7)$$\frac{dC_{a}}{dz}=\frac{-W}{Q_{a}}\left[K_{m}\left(C_{1}-\frac{{K^{\prime }}_{ex}C_{2}{}_{\left(\ell \right)}}{H_{c}}\right)\right]=\frac{-W}{Q_{a}}\left[K_{m}\gamma_{m}\left(C_{a}-\frac{C_{b(\ell )}}{H_{c}}\right)\right]$$
(8a)$$\frac{dC_{b}}{dz}=\frac{-k_{CO_{2}}C_{b(\ell )}(WH)}{Q_{b}}+\frac{W}{Q_{b}}\left[K_{m}\left(C_{1}-\frac{{K^{\prime }}_{ex}C_{2}{}_{\left(\ell \right)}}{H_{c}}\right)\right]=\frac{-k_{CO_{2}}C_{b(\ell )}(WH)}{q_{b}}+\frac{W}{q_{b}}\left[K_{m}\gamma_{m}\left(C_{a}-\frac{C_{b(\ell )}}{H_{c}}\right)\;\right]\;=\frac{-k_{CO_{2}}C_{b(\ell )}(WH)}{q_{b}}+\frac{W}{q_{b}}\left[K_{m}\gamma_{m}\left(C_{a}-\frac{C_{b(\ell )}}{H_{c}}\right)\;\right],\;\mathrm{concurrent}\;\mathrm{flow}$$
(8b)$$\frac{dC_{b}}{dz}=\frac{k_{CO_{2}}C_{b(\ell )}(WH)}{Q_{b}}-\frac{W}{Q_{b}}\left[K_{m}\left(C_{1}-\frac{{K^{\prime }}_{ex}C_{2}{}_{\left(\ell \right)}}{H_{c}}\right)\right]\;=\frac{k_{CO_{2}}C_{b(\ell )}(WH)}{Q_{b}}-\frac{W}{Q_{b}}\left[K_{m}\gamma_{m}\left(C_{a}-\frac{C_{b(\ell )}}{H_{c}}\right)\;\right],\;\mathrm{countercurrent}\;\mathrm{flow}$$
in which *z* is the coordinate along with the flowing direction, and the concentration polarization coefficient $\gamma_{m}$ was derived and obtained by equating Equations (1) and (3) ($J_{m}=J_{g}$) and Equations (2) and (3) ($J_{m}=J_{\ell }$), respectively, as follows:(9)$$\gamma_{m}=\frac{\left(C_{1}-\frac{{K^{\prime }}_{ex}C_{2\left(\ell \right)}}{H_{c}}\right)}{\left(C_{a}-\frac{C_{b\left(\ell \right)}}{H_{c}}\right)}=\frac{k_{a}k_{b}}{k_{a}k_{b}+k_{m}k_{a}+k_{m}k_{b}}$$

The fourth-order Runge–Kutta method is a good approximation that lumps four sampled slopes to obtain the dependent variables with the minimum tradeoff between accuracy and computational results. The accumulated roundoff error of the fourth-order Runge–Kutta method may be reduced substantially by using the fourth-order Runge–Kutta numerical scheme. The simultaneous ordinary equations of Equations (7) and (8a) for concurrent flow operation and Equations (7) and (8b) for countercurrent flow operation) in Figure 7a,b, respectively, were solved with the use of the estimated convective heat-transfer coefficients and calculated iteratively in Figure 8 and Figure 9 by marching the fourth-order Runge–Kutta method numerically along the membrane absorption module, and the CO_2_ absorption flux and absorption flux improvement were obtained accordingly. The concentration distributions were predicted theoretically not only in the gas/liquid bulk flows but also in the membrane surfaces of both gas/liquid feed streams under concurrent and countercurrent flow operations, respectively. Comparisons were made between the module with embedded 3D turbulence promoters and with empty channels. An iterative procedure, as illustrated in Figure 8, was used to calculate the CO_2_ absorption flux $J_{theo}$ for concurrent flow operation, whereas an additional guess of CO_2_ concentration at the inlet of the MEA feed $C_{b,j=n}$ needed to be specified for the countercurrent flow operation calculation, as illustrated in Figure 9.

The 3D turbulence promoters were embedded in the MEA feed stream instead of using the module of the empty channel (without embedding turbulence promoters). The extent of absorption flux increment is incorporated into an enhancement factor [31], which is the ratio of the Sherwood number of the module with embedded 3D turbulence promoters to that of the module using empty channels. The mass-transfer enhancement factor $\alpha^{S}$ depending on various geometric shapes, array patterns of 3D turbulence promoters, and flow patterns was correlated to estimate the augmented mass-transfer coefficients in the gas/liquid membrane contactors as follows:(10)$$Sh^{P}=\frac{k_{b}D_{h}{}_{,spriral}}{D_{b}}=\alpha^{P}Sh_{lam}$$

The correlated equation [39] for the membrane contactor module using empty channels under laminar flow is:(11)$$Sh_{lam}=0.023\;Re^{0.8}Sc^{0.33}$$

The Sherwood number of embedding 3D turbulence promoters into the flow channel can be lumped into four dimensionless groups using Buckingham’s π theorem:(12)$$Sh^{P}=f(\frac{D_{h,promoter}}{D_{h,empty}},Re,Sc)$$
where $D_{h,promoter}$ and $D_{h,empty}$ are the equivalent diameters of modules with embedded 3D turbulence promoters and empty channels, respectively.

The energy consumption increment due to the increased frictional loss caused by embedding 3D turbulence promoters in the flow channel of the double-unit parallel-plate membrane contactor module was determined using the Fanning friction factor $f_{F}$ for both laminar and turbulent flows [40]:(13)$$H_{i}=Q_{a}\;\rho_{CO{\;}_{2}}\;\ell w_{f,CO{\;}_{2}}+Q_{b}\;\rho_{MEA}\;\ell w_{f,MEA}\;i=promoter,empty$$
(14)$$\ell w_{f,j}=\frac{2f_{F,j}{\bar{v}}_{j}^{2}L}{D_{h,i}},\;j=CO_{2},MEA$$

The average velocity and equivalent hydraulic diameter of each flow channel were calculated as follows:(15)$${\bar{\nu }}_{CO_{2}}=\frac{Q_{a}}{WH},\;{\bar{\nu }}_{MEA}=\frac{Q_{b}}{H(W-W_{1}N_{1})}$$
(16)$$D_{h,empty}=\frac{4HW}{2(H+W)},\;D_{h,promoter}=\frac{4H(W-W_{1}N_{1})}{2(W+H+D_{1}N_{1})}$$

The relative extents $I_{H}$ of the power consumption increment were illustrated by calculating the percent increase in the device with embedded 3D turbulence promoters, which was based on the device with an empty channel (wound with nylon fiber):(17)$$I_{H}=\frac{H_{promoter}-H_{empty}}{H_{empty}}\times 100\%$$
where the subscripts *promoter* and *empty* represent the flow channels with and without embedded 3D turbulence promoters, respectively.

## 4. Results and Discussions

The CO_2_ absorption flux for various MEA feed flow rates, inlet feed CO_2_ concentrations, and turbulence promoter configurations were obtained using numerical Runge–Kutta marching scheme with Equations (7) and (8a) and Equations (7) and (8b) for concurrent and countercurrent flow operations, respectively, and hence, comparisons were made on device performance for both empty and promoter modules, as indicated in Figure 10.

Good agreement of the theoretical predictions with those obtained from experimental results was achieved. Two geometric shapes and two array patterns of the embedded 3D turbulence promoters produce higher turbulence intensity, which results in a higher mass transfer or higher absorption flux. The results showed the CO_2_ absorption flux for the module with the embedded 3D turbulence promoter with geometric shapes of Circle and Square in both concurrent and countercurrent flow operations. In general, the module embedding the 3D turbulence promoter showed a more significant CO_2_ transporting flux through the hydrophobic membrane in countercurrent flow operations than that in concurrent flow operations because of a larger concentration driving-force gradient.

The absorption flux in the device with embedded 3D turbulence promoters was presented graphically, as delineated in Figure 11, Figure 12, Figure 13 and Figure 14, for Circle Type A, Circle Type B, Square Type A, and Square Type B, respectively, with the geometric shape and flow pattern as parameters. The order of the theoretical performance of absorption flux for the device embedding 3D turbulence promoters was as follows: Square Type B > Circle Type B > Circle Type A > Square Type A. As expected, and as seen in Figure 11, Figure 12, Figure 13 and Figure 14, increases in both the MEA feed flow rate and the inlet feed CO_2_ concentration yielded higher absorption fluxes.

The accuracy deviation [41] of the experimental results from the theoretical predictions was calculated using the following definition:(18)$$Er\;(\%)=\frac{1}{N_{\exp}}\sum_{i=1}^{N_{\exp}}\frac{\left|J_{theo,i}-J_{\exp,i}\right|}{J_{\exp,i}}\times 100$$
where *N_exp_*, $J_{\exp,i}$ and $J_{theo,i}$ are the number of experimental runs, theoretical predictions, and experimental results of the absorption fluxes, respectively. The accuracy deviations with two flow patterns were calculated; the agreement of experimental results deviated from the theoretical predictions was good, within $3.0\times 10^{-4}\leq Er\leq 1.9\times 10^{-2}$.

The present work extends the previous study, except it embedded 3D turbulence promoters instead of carbon-fiber spacers [25]. To perform additional membrane absorption tests, the experiment runs were conducted on the channels of the membrane absorption modules with 3D turbulence promoters to replace the carbon-fiber spacers, as shown in Figure 15. The present study of 3D turbulence promoters and carbon-fiber spacers [25] illustrates why the present design of embedding 3D turbulence promoters is preferred, as shown in Figure 15 for both concurrent and countercurrent flow operations. The results also showed a considerably larger chemical absorption in MEA solution compared with our previous work [10] that used water as an absorbent. This is the value and originality of the present study regarding the technical feasibility.

The absorption flux improvement $I_{p}$ was illustrated by calculating the percentage increase in the device with embedded 3D turbulence promoters on the basis of the device of the empty channel (wound with nylon fiber) as follows:(19)$$I_{empty}^{CT}\left(\%\right)=\frac{J_{empty}^{CT}-J_{empty}^{CN}}{J_{empty}^{CN}}\times 100=\left(\frac{J_{empty}^{CT}}{J_{empty}^{CN}}-1\right)\times 100$$
(20)$$I_{P}^{CN}\left(\%\right)=\frac{J_{P}^{CN}-J_{empty}^{CN}}{J_{empty}^{CN}}\times 100=\left(\frac{J_{P}^{CN}}{J_{empty}^{CN}}-1\right)\times 100$$
(21)$$I_{P}^{CT}\left(\%\right)=\frac{J_{P}^{CT}-J_{empty}^{CN}}{J_{empty}^{CN}}\times 100=\left(\frac{J_{P}^{CT}}{J_{empty}^{CN}}-1\right)\times 100$$
where $I_{empty}^{CT}$, $I_{P}^{CN}$ and $I_{P}^{CT}$ are the absorption flux improvement for countercurrent-flow operations with empty channel and concurrent and countercurrent flow operations with embedded 3D turbulence promoters, respectively. Meanwhile, the subscripts *P* and *empty* denote the MEA flow channels with/without embedded 3D turbulence promoters, respectively, and the superscripts *CN* and *CT* denote concurrent and countercurrent flow operations, respectively. The percentage increase in absorption flux improvement $I_{p}$ was evaluated in the comparisons of the absorption flux in the module with the embedded 3D turbulence promoter to that of the empty channel, as seen in Figure 16 for both the Circle and Square turbulence promoters under two flow patterns, respectively. The theoretical predictions show that an absorption flux improvement of up to 40% was obtained with embedded Square turbulence promoters of Type B array patterns compared with that in the empty channel device, as seen in Figure 16. 

Meanwhile, the CO_2_ absorption flux augmented by embedding 3D turbulence promoters is more considerable in countercurrent-flow operations than that in concurrent-flow operations. The Square turbulence promoter with Type B array patterns enhances the absorption flux enhancement by approximately 15% compared to Type A array patterns, whereas the Circle turbulence promoters of Type A shows a lower absorption flux enhancement than that in Type B by approximately 5% under the same operating conditions. The absorption flux improvement increases with inlet feed CO_2_ concentration but decreases with MEA feed flow rate, as shown in Figure 16. Generally, embedding 3D turbulence promoters into the flow channel shows a significant influence to increase the absorption flux in the double-unit flat-plate gas/liquid membrane contactor module.

The concentration polarization coefficient $\gamma_{m}$ is an indicator of the mass transfer resistance, which was defined in Equation (9) and calculated by CO_2_ concentration distributions, as illustrated in Table 1, with various MEA feed flow rates and inlet feed CO_2_ concentration as parameters. The magnitude of the concentration polarization coefficient $\gamma_{m}$ was governed by the concentration boundary layer in both gas/liquid feed streams, especially on the membrane surface in the MEA bulk flow. The theoretical predictions of the concentration polarization coefficient $\gamma_{m}$ show that the value of $\gamma_{m}$ increased with increases in the MEA feed flow rates but with decreases in inlet feed CO_2_ concentrations. The smaller $\gamma_{m}$ deviated from unity and the greater polarization effect dominated, which was diminished by the higher MEA feed flow rate as well as the higher eddy promotion produced by embedding 3D turbulence promoters into the flow channel. The absorption flux improvement was enhanced by embedding the turbulence promoter; a positive influence on the shrinking concentration of polarization layers was observed, for which a higher $\gamma_{m}$ value and a higher absorption flux improvement are expected. In addition, a larger $\gamma_{m}$ value (a lesser mass transfer resistance) was achieved in the countercurrent flow operation than that in the concurrent flow operation. The mass transfer resistance dominating the CO_2_ absorption flux decreased with an increase in the inlet feed CO_2_ concentration and the MEA feed flow rate. The influence of concentration polarization coefficients $\gamma_{m}$ on mass transfer behavior was confirmed in both concurrent and countercurrent flow operations, as shown in Table 1. Regarding the influences of the geometric shapes and configurations of turbulence promoters on the absorption efficiency, the relative increments of the concentration polarization coefficient $\gamma_{m}$ with respect to the MEA feed flow rates and inlet feed CO_2_ concentrations were more significant in the concurrent flow operations than those in the countercurrent flow operations. Meanwhile, the Type A configuration and Circle turbulence promoters show more effective increments of the concentration polarization coefficient $\gamma_{m}$ than those from the modules with a Type B configuration and Square turbulence promoters, respectively.

The further absorption flux enhancement $E_{p}$ of CO_2_ absorption in membrane contactors by embedding 3D turbulence promoters in the flow channel was calculated on the basis of the device with the same working dimensions as that of the device under countercurrent flow operations, as follows:$$E_{P}\left(\%\right)=\frac{J_{P}^{CT}-J_{empty}^{CT}}{J_{empty}^{CT}}\times 100=\left[\frac{\left(J_{P}^{CT}-J_{empty}^{CN})-(J_{empty}^{CT}-J_{empty}^{CN}\right)}{J_{empty}^{CN}}\right]\left(\frac{J_{empty}^{CN}}{J_{empty}^{CT}}\right)\times 100$$
(22)$$=\left(I_{P}^{CT}-I_{empty}^{CT}\right)\;\left(\frac{J_{empty}^{CN}}{J_{empty}^{CT}}\right)\times 100=\left(I_{P}^{CT}-I_{empty}^{CT}\right)/\left(1+I_{empty}^{CT}\right)\times 100$$

Further absorption flux enhancement was accomplished if there were various geometric shapes of turbulence promoters embedded into the MEA feed stream under Type A and Type B array patterns and flow patterns. Generally, the further absorption flux enhancement of the module with an embedded 3D turbulence promoter increased with an increase in the inlet feed CO_2_ concentration but decreased with the MEA feed flow rate, as indicated in Table 2.

The mass transfer coefficients determined by the theoretical model and expressed in terms of Sherwood number (the correlated Sherwood numbers) in comparison with the experimental data in the device with embedded 3D turbulence promoters, was obtained by Equation (12) and presented graphically in Figure 17 as follows:(23)αP=ShPShlam=0.218 lnDh,promoterDh, empty−0.296Re0.448

The normal equations for the least square parameters were set up to find the fitting function in a linear model, for which the regression line was determined to be linear and hence easily solved. The correlated Sherwood numbers, as shown in Figure 17, indicate that the mass transfer coefficient of the device with embedded Square turbulence promoters in the Type B array pattern achieved a higher value than those of the devices with Square Type A as well as with embedded Circle turbulence promoters. Embedding turbulence promoters played an important role in the absorption flux improvement because of the disruption of the concentration boundary layer with mass-transfer resistance reduction, and thus the CO_2_ absorption flux was enhanced because a higher turbulence intensity was induced. In other words, the non-smooth curvature geometric shape of Square turbulence promoters embedded in flow channels created higher turbulence vortices and resulted in a larger absorption flux.

An economic viewpoint in making a suitable selection was examined for both desirable absorption flux improvement and undesirable power consumption increase due to embedding turbulence promoters into the flow channel. Concerning the compensation of the CO_2_ absorption flux improvement accompanied with the friction loss increase by embedding turbulence promoters in the MEA feed channel, the effects of geometric shapes of 3D turbulence promoters, configurations, inlet feed CO_2_ concentrations, flow patterns, and MEA flow rates are shown in Figure 18, referring to the ratio of $I_{P}/I_{H}$. Figure 18 shows that the countercurrent flow operation obtained a higher CO_2_ absorption flux than that of the concurrent flow operation under the same the friction loss increment, and thus a relatively larger $I_{P}/I_{H}$ value was achieved in the countercurrent flow operation with respect to the economic consideration.

We also found that the helpfulness of embedding turbulence promoters with Type B array pattern was higher than that of Type A array pattern under both flow operations. Meanwhile, the increase in the MEA feed flow rate yielded a lower ratio of $I_{P}/I_{H}$ and reached a steep decrease for the higher MEA feed flow rate, being larger than $8.33\times 10^{-6}\;$m^3^/s. The order of the ratio of $I_{P}/I_{H}$ is expected to have the same trend of the absorption fluxes, as follows: Square Type B > Circle Type B > Circle Type A > Square Type A.

## 5. Conclusions

Two geometric shapes of turbulence promoters were used for gas/liquid membrane contactors and compared to an empty channel under two flow patterns in the present study. The CO_2_ absorption flux in the MEA solution increased with an increase in both the MEA feed flow rate and the inlet feed CO_2_ concentration. For empty channels, the CO_2_ absorption flux had no obvious change with changes in the inlet feed CO_2_ concentration. Increasing shear stress on the membrane surface due to embedding turbulence promoters could effectively reduce the concentration polarization effect in the concentration boundary layer. In Square Type B, compared with the empty-channel type, the absorption flux improvement increased by approximately 37% and 40% for higher inlet feed CO_2_ concentration under concurrent and countercurrent flow operations, respectively. Compared with the empty channel type, the absorption flux enhancement of the Circle type was greater than that of the normal type by approximately 17–26% and 20–32% for various inlet feed CO_2_ concentrations for Type A and Type B, respectively. The comparisons of the absorption flux enhancement with the embedding of various geometric shapes of 3D turbulence promoters on the CO_2_ absorption in MEA absorbent of double-unit flat-plate membrane absorption modules led to the following conclusions:The absorption flux enhancement increases with an increase in the volumetric flow rate.The higher the inlet saline temperature yields a higher absorption flux enhancement.The absorption flux enhancement is obtained by embedding both Circle and Square shapes of 3D turbulence promoters, and the improvement of the Type B configuration is higher than that of Type the A configuration.A more considerable absorption flux is accomplished in countercurrent flow operations than that in concurrent flow operations because of the larger concentration gradient across both membrane surfaces.A maximum of 40% absorption flux enhancement was found in the module with embedding Square turbulence promoters of the Type B configuration compared with that in the empty-channel module under the countercurrent-flow operation.The economic viewpoint of $I_{E}/I_{P}$ for absorption flux enhancement to power consumption increment indicates that the energy utilization is more effective for the module with embedding 3D turbulence promoters at the higher MEA flow rate.The ratio of $I_{E}/I_{P}$ for the Type B configuration is higher than that of the Type A configuration.

The new design in this study includes the advantageous effect of strengthening the turbulence intensity as an alternative strategy [25] on the absorption flux in a double-unit flat-plate membrane absorption module. The value of this membrane absorption module is easier to implement the experimental apparatus with a lower fabricating cost. The results demonstrated its technical and economic feasibility in terms of the ratio of $I_{E}$ to $I_{P}$ by embedding turbulence promoters in the flowing channel. Overall, adding the Square turbulence promoter to the gas/liquid membrane contactor module on the CO_2_ absorption in MEA solution shows great potential to considerably diminish the concentration polarization effect and enhance the absorption flux. Furthermore, a simplified expression of the Sherwood number was obtained to correlate the mass transfer coefficient of the gas/liquid membrane contactor module with embedded 3D turbulence promoters in the flow channel.

## Figures and Tables

**Figure 1 membranes-12-00370-f001:**
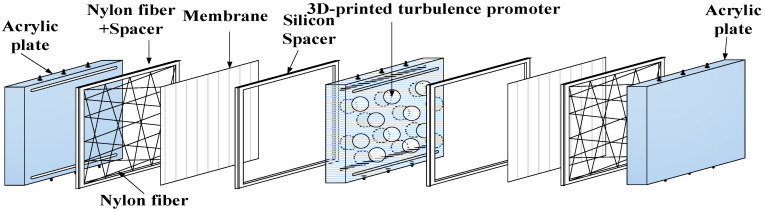
Schematic diagram of the double-unit flat-plate membrane contactor module.

**Figure 2 membranes-12-00370-f002:**
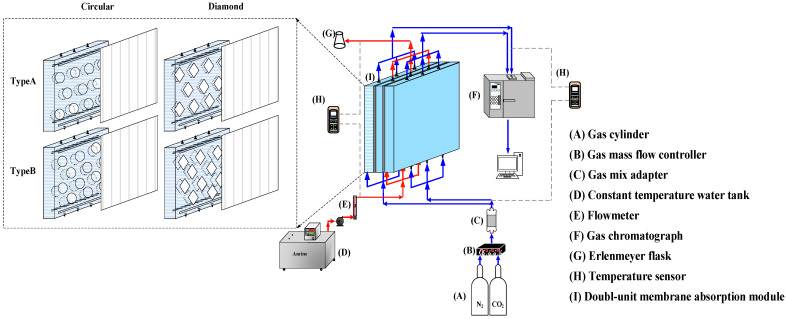
Schematic representation of the double-unit flat-plate membrane contactor.

**Figure 3 membranes-12-00370-f003:**
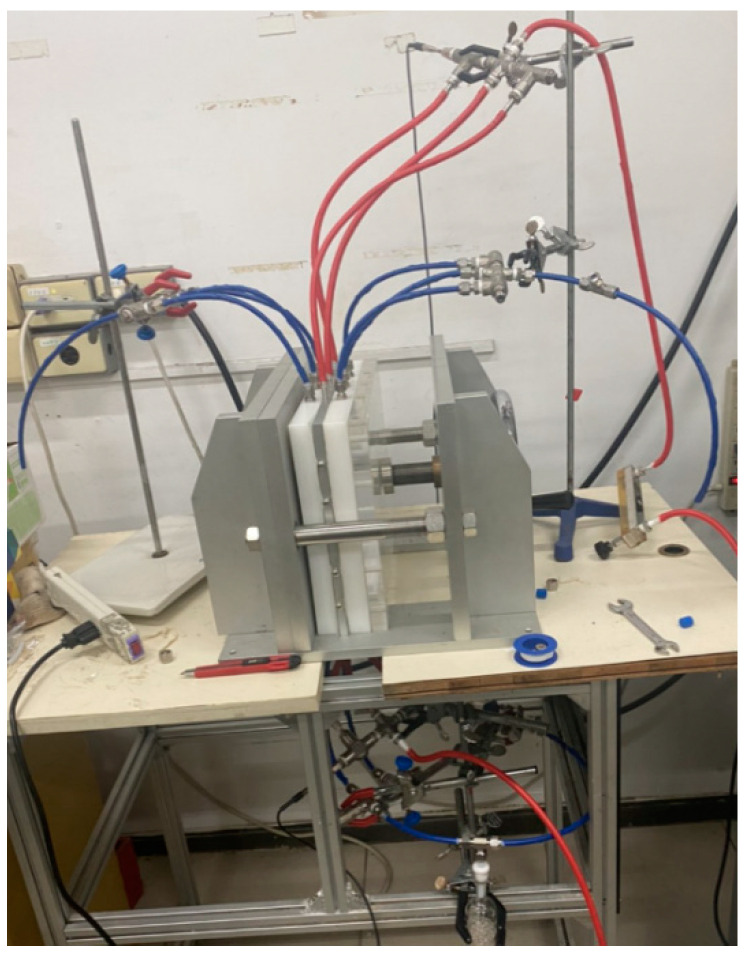
A photo of the experimental apparatus of the double-unit flat-plate membrane contactor.

**Figure 4 membranes-12-00370-f004:**
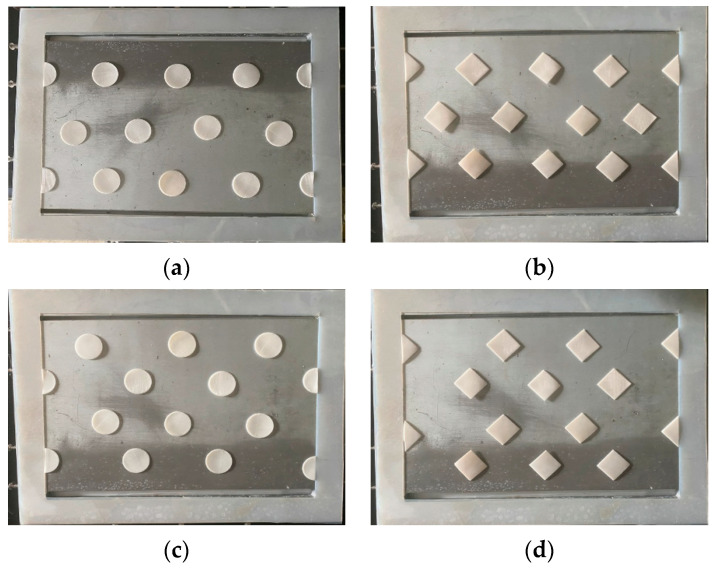
Top view of two shapes and two configurations of 3D turbulence promoters. (**a**) Circle shape (Type A), (**b**) Square shape (Type A), (**c**) Circle shape (Type B), (**d**) Square shape (Type B).

**Figure 5 membranes-12-00370-f005:**
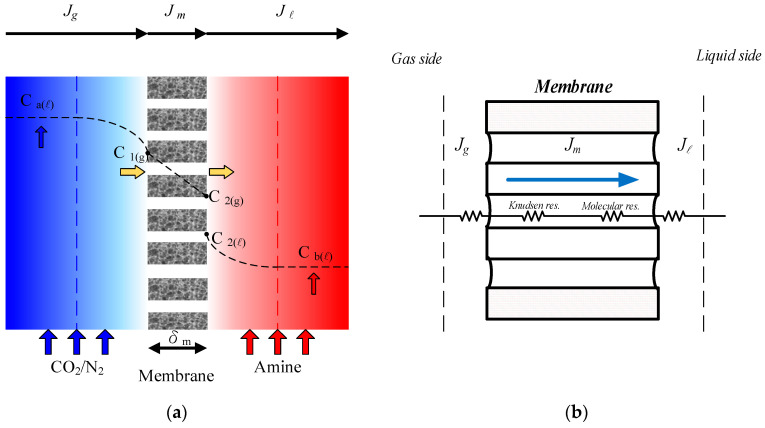
Schematic mass transfer resistances and concentration profiles of membrane contactor. (**a**) CO_2_ concentration variations, (**b**) Mass transfer resistances.

**Figure 6 membranes-12-00370-f006:**
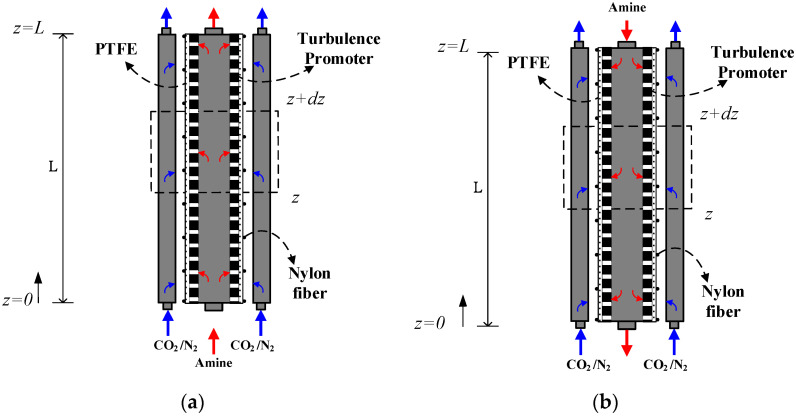
The mass balance made within a finite fluid element. (**a**) Concurrent flow operations, (**b**) countercurrent flow operations.

**Figure 7 membranes-12-00370-f007:**
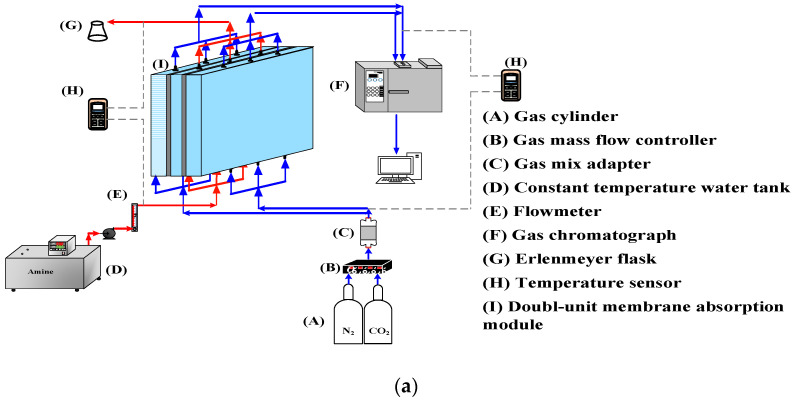
Schematic diagram of double-unit flat-plate membrane contactor modules. (**a**) Concurrent flow operations, (**b**) countercurrent flow operations.

**Figure 8 membranes-12-00370-f008:**
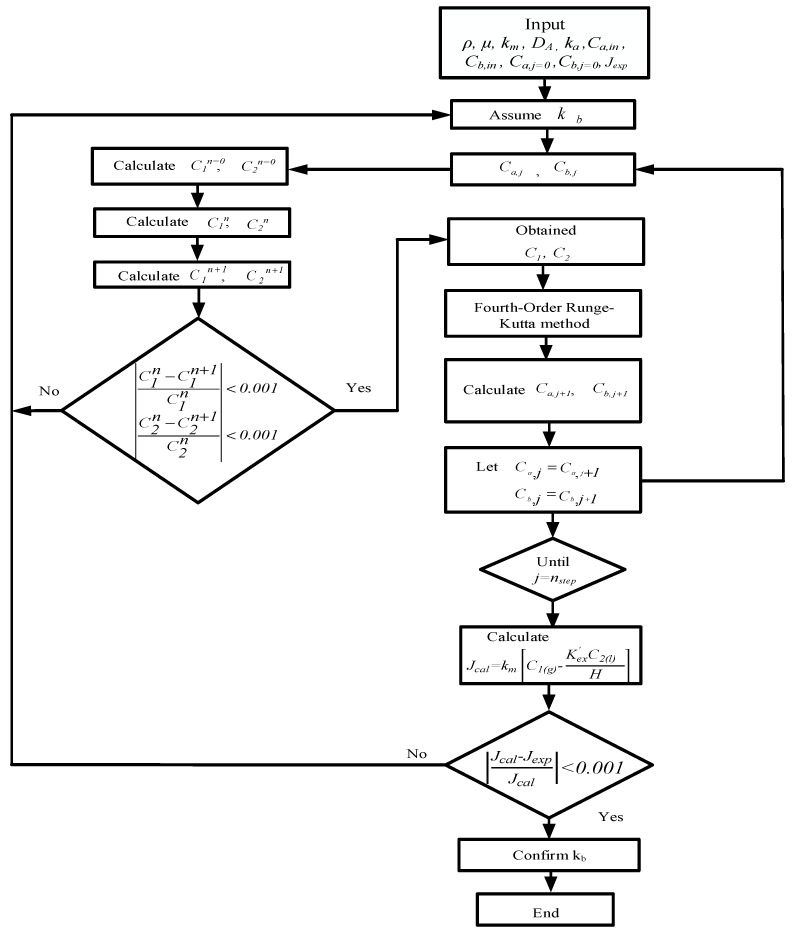
Calculation flow chart for determining CO_2_ concentrations in gas and liquid phases under concurrent flow operations.

**Figure 9 membranes-12-00370-f009:**
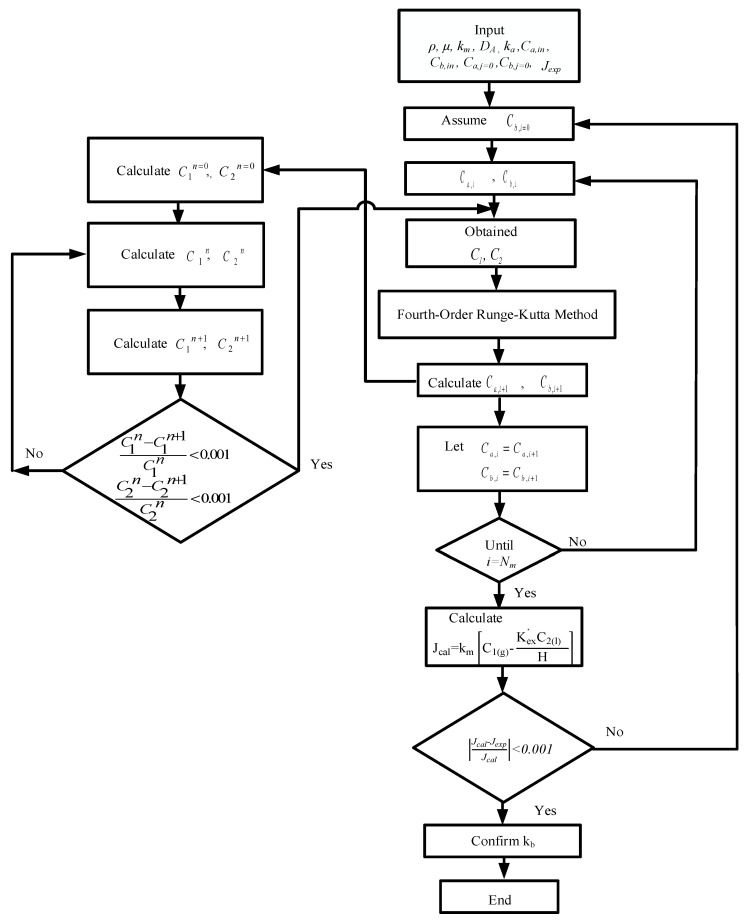
Calculation flow chart for determining CO_2_ concentrations in gas and liquid phases under countercurrent flow operations.

**Figure 10 membranes-12-00370-f010:**
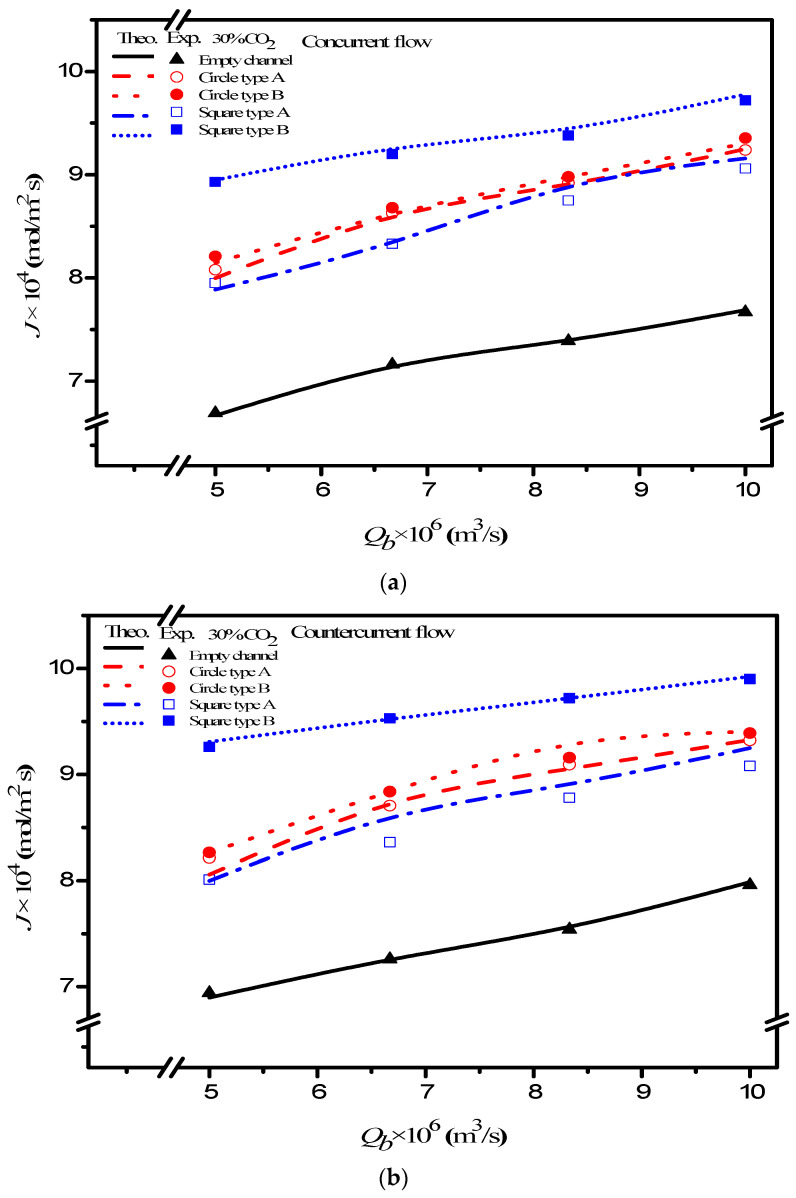
Effects of MEA flow rate and configurations of turbulence promoters on absorption rate. (**a**) Concurrent flow operations, (**b**) countercurrent flow operations.

**Figure 11 membranes-12-00370-f011:**
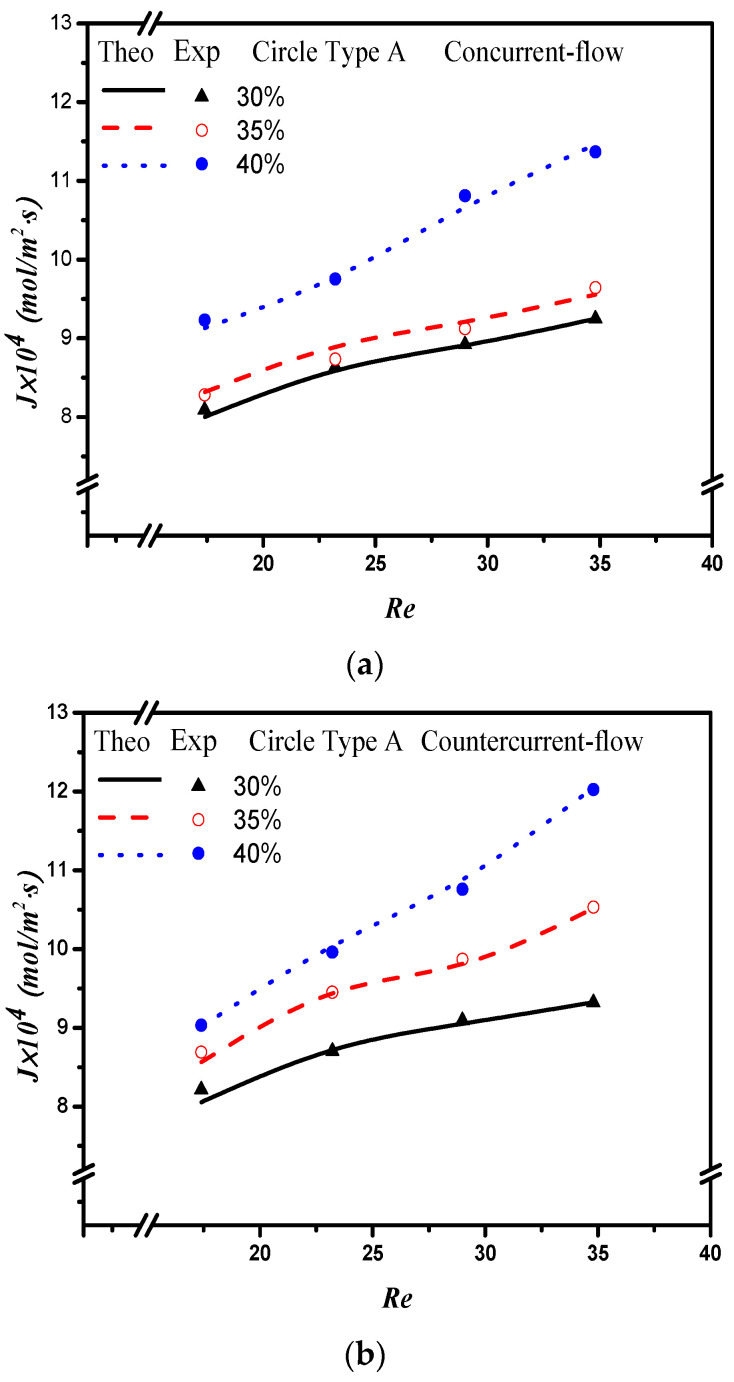
Effects of inlet CO_2_ feed concentrations and flow patterns for the CO_2_ absorption flux. (**a**) Concurrent flow operations, (**b**) countercurrent flow operations.

**Figure 12 membranes-12-00370-f012:**
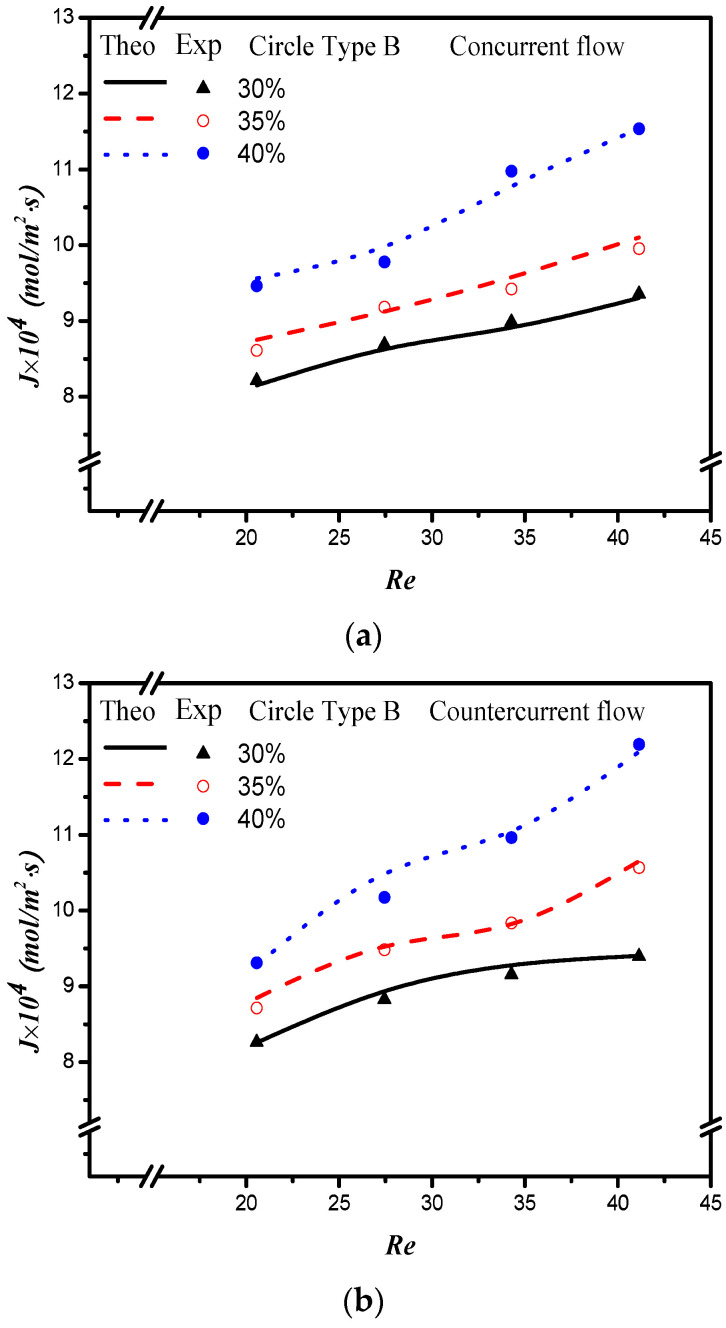
Effects of inlet CO_2_ feed concentrations and flow patterns for the CO_2_ absorption flux. (**a**) Concurrent flow operations, (**b**) countercurrent flow operations.

**Figure 13 membranes-12-00370-f013:**
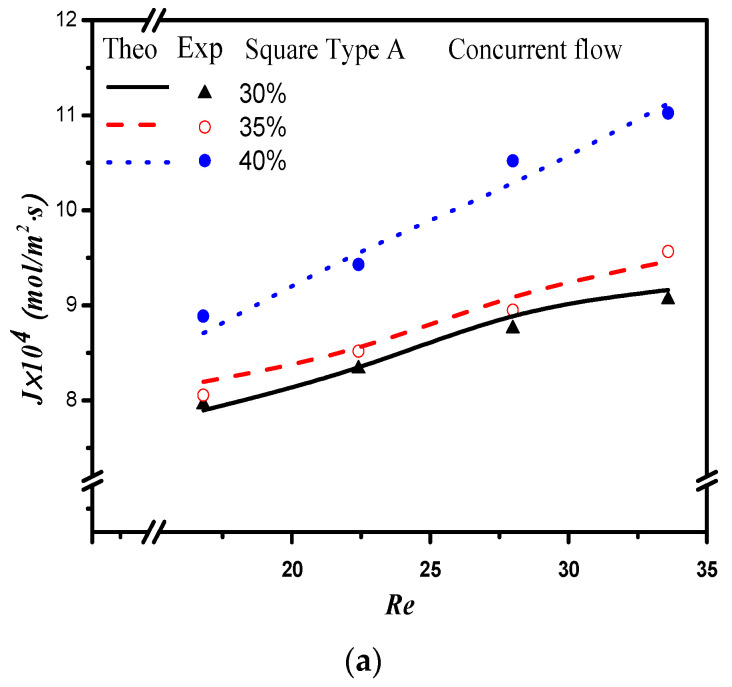
Effects of inlet CO_2_ feed concentrations and flow patterns for the CO_2_ absorption flux. (**a**) Concurrent flow operations, (**b**) countercurrent flow operations.

**Figure 14 membranes-12-00370-f014:**
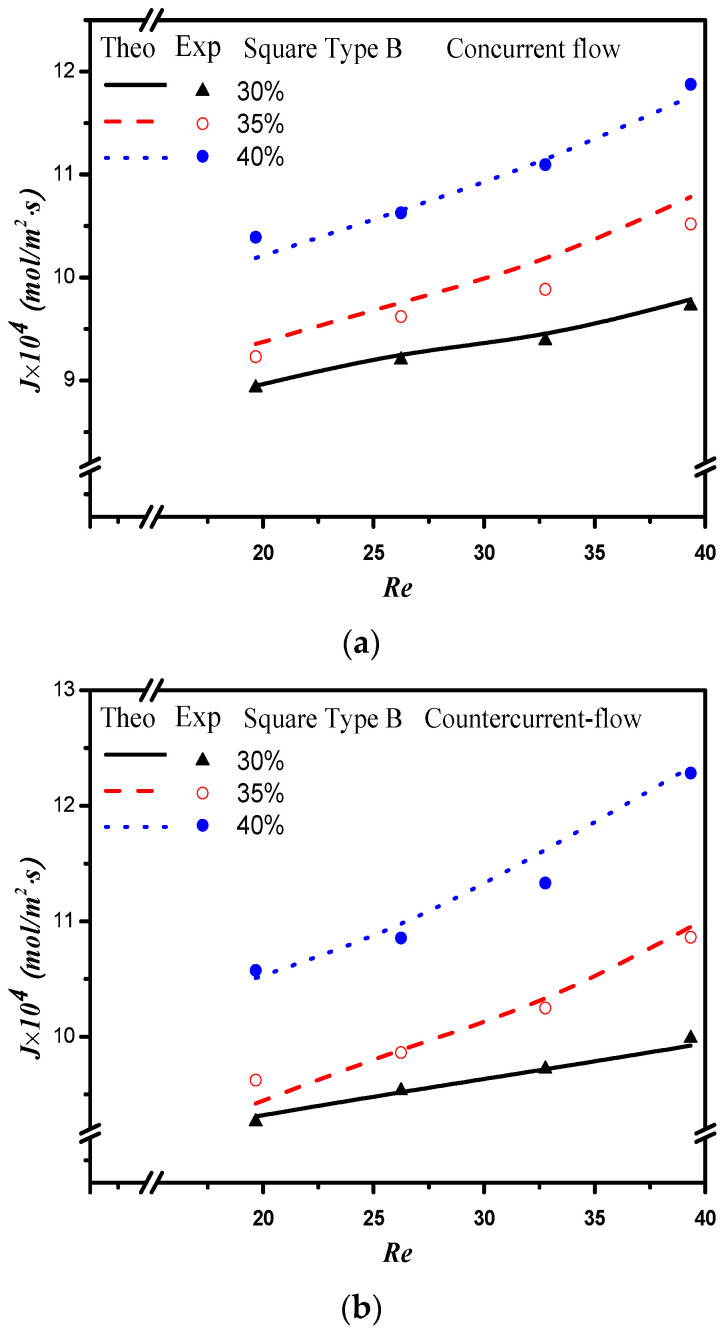
Effects of inlet CO_2_ feed concentrations and flow patterns for the CO_2_ absorption flux. (**a**) Concurrent flow operations, (**b**) countercurrent flow operations.

**Figure 15 membranes-12-00370-f015:**
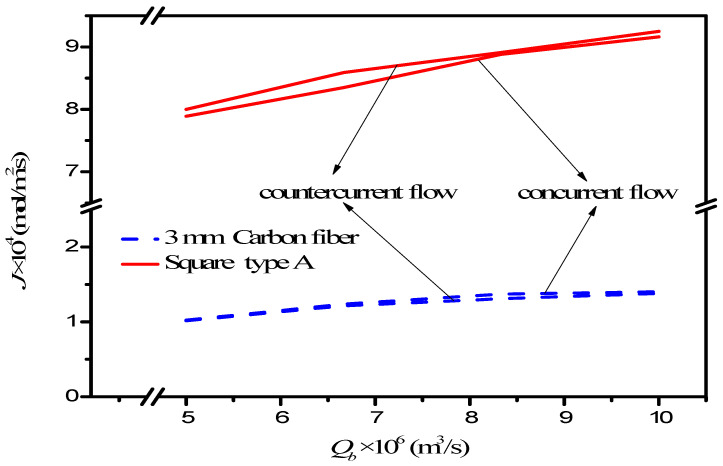
Comparisons of theoretical CO_2_ absorption flux of the channels with embedded 3D turbulence promoters and inserted carbon-fiber spacers.

**Figure 16 membranes-12-00370-f016:**
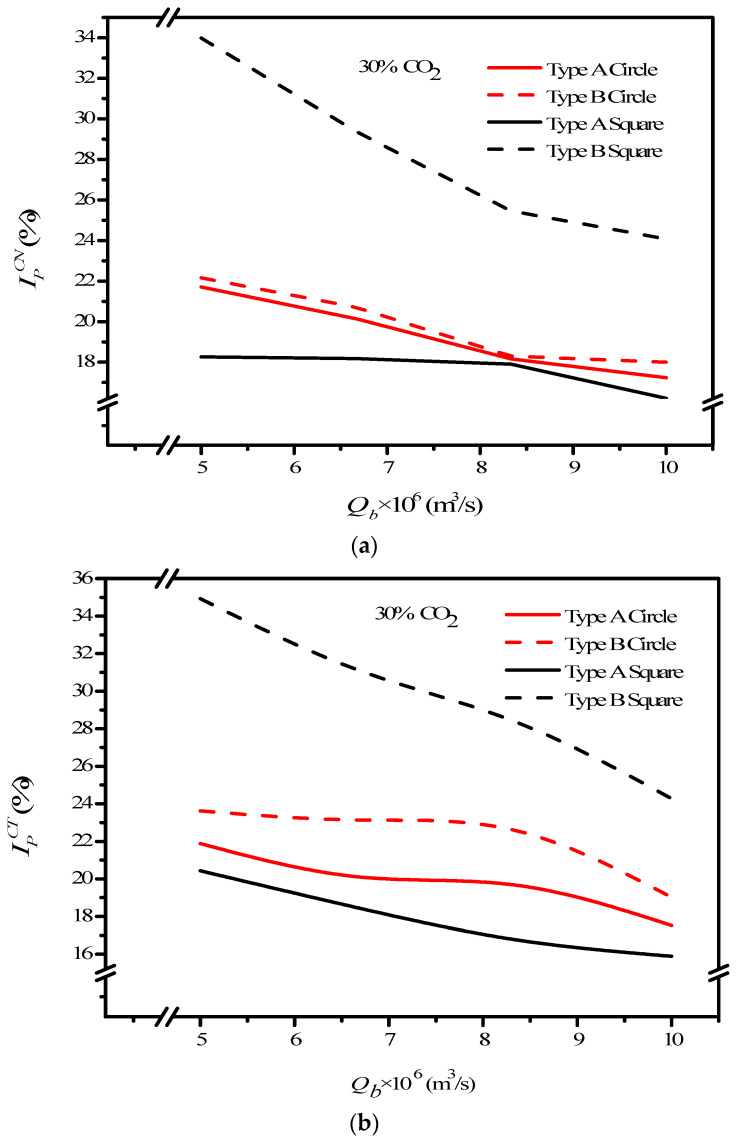
Effects of shapes of turbulence promoters on absorption flux improvements. (**a**) Concurrent flow operations, (**b**) countercurrent flow operations.

**Figure 17 membranes-12-00370-f017:**
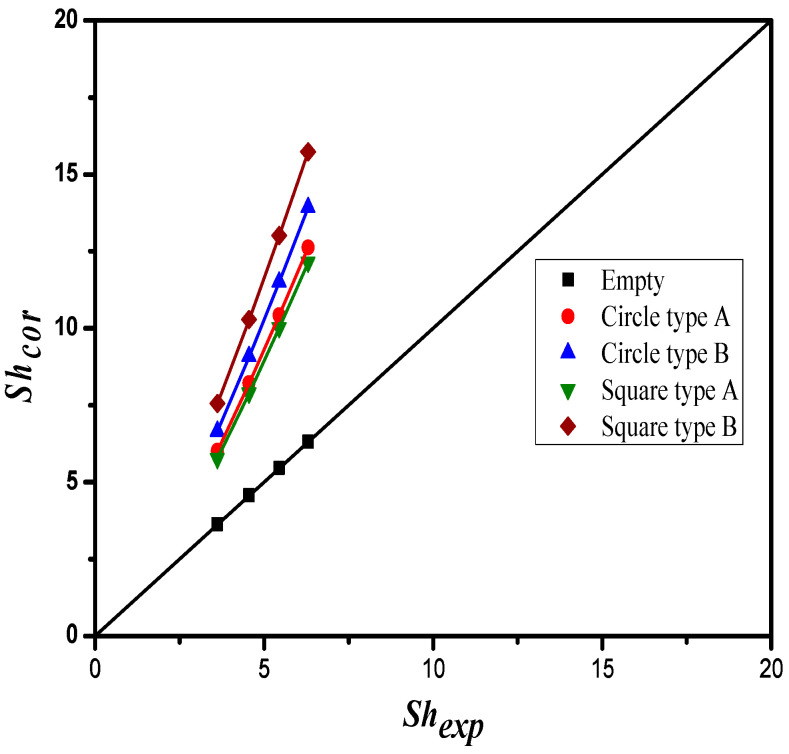
Comparison of correlated and experimental Sherwood numbers for empty channel and channels with embedding 3D turbulence promoters under various array patterns.

**Figure 18 membranes-12-00370-f018:**
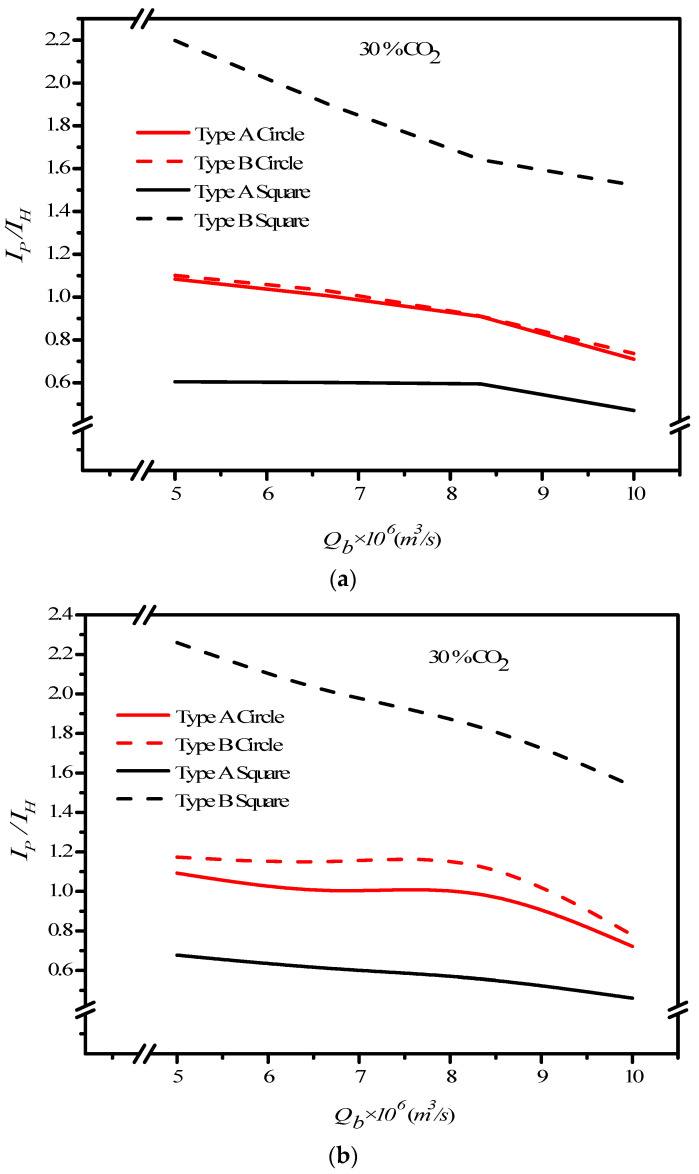
Effects of MEA feed flow rate and promoter shapes on $I_{P}/I_{H}$. (**a**) Concurrent flow operations, (**b**) countercurrent flow operations.

**Table 1 membranes-12-00370-t001:** Effects of shapes of turbulence promoters on concentration polarization coefficients $\gamma_{m}$.

$\begin{array}{c} C_{in}\;Q_{b}\times 10^{6}(\%)\\ \left(m^{3}/s\right) \end{array}$		Concurrent Flow Operations				Countercurrent Flow Operations			
		Type A		Type B		Type A		Type B	
		Circle	Square	Circle	Square	Circle	Square	Circle	Square
30	5.00	0.383	0.367	0.397	0.432	0.417	0.408	0.428	0.436
	6.67	0.434	0.402	0.443	0.441	0.426	0.419	0.453	0.464
	8.33	0.448	0.428	0.458	0.455	0.462	0.425	0.459	0.459
	10.0	0.470	0.448	0.465	0.482	0.472	0.464	0.469	0.499
35	5.00	0.379	0.359	0.388	0.409	0.400	0.383	0.400	0.416
	6.67	0.411	0.389	0.422	0.440	0.411	0.400	0.424	0.443
	8.33	0.437	0.407	0.434	0.441	0.443	0.423	0.448	0.449
	10.0	0.452	0.432	0.443	0.462	0.459	0.452	0.451	0.486
40	5.00	0.378	0.338	0.381	0.388	0.391	0.344	0.379	0.376
	6.67	0.390	0.378	0.395	0.420	0.399	0.366	0.410	0.415
	8.33	0.426	0.387	0.425	0.429	0.426	0.419	0.424	0.437
	10.0	0.438	0.408	0.415	0.449	0.442	0.443	0.438	0.477

**Table 2 membranes-12-00370-t002:** Theoretical predictions of further absorption flux enhancement $E_{P}$.

$\begin{array}{c} C_{in}\;Q_{b}\times 10^{6}(\%)\\ \left(m^{3}/s\right) \end{array}$		Countercurrent Flow Operations								
		**Empty** **Channel**	Circle				Square			
			**Type A**		Type B		Type A		Type B	
		$I_{empty}^{CT}$ (%)	$I_{P}^{CT}$ (%)	$E_{P}$ (%)	$I_{P}^{CT}$ (%)	$E_{P}$ (%)	$I_{P}^{CT}$ (%)	$E_{P}$ (%)	$I_{P}^{CT}$ (%)	$E_{P}$ (%)
30	5.00	3.29	21.88	18.00	23.62	19.68	20.44	16.41	34.93	30.63
	6.67	1.54	20.11	18.29	23.14	21.27	18.46	16.66	31.13	29.14
	8.33	0.40	19.68	19.20	22.59	22.10	16.77	16.30	28.40	27.89
	10.0	1.27	17.52	15.31	19.02	16.29	15.89	14.44	24.28	22.72
35	5.00	4.35	24.58	19.39	25.42	20.19	23.06	17.93	35.83	29.66
	6.67	3.50	22.92	18.76	24.09	19.89	20.97	16.88	31.64	27.19
	8.33	3.30	21.35	17.47	22.95	19.02	20.12	16.28	28.72	24.61
	10.0	3.58	21.06	16. 88	22.67	18.43	18.87	14.76	26.01	21.66
40	5.00	1.22	26.70	24.26	32.04	30.45	25.23	23.72	40.32	38.63
	6.67	1.14	25.81	24.39	31.45	29.97	24.19	22.79	37.59	36.04
	8.33	5.98	22.91	15.98	29.80	22.48	23.93	16.94	32.85	25.35
	10.0	5.62	21.94	15.45	28.86	22.00	23.43	16.87	31.42	24.43

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
