# Peer review of "Theoretical and Experimental Studies of CO2 Absorption in Double-Unit Flat-Plate Membrane Contactors"

_membranes, 2022, doi:10.3390/membranes12040370_

Round 1
Reviewer 1 Report
The present work deals Theoretical predictions of the carbon dioxide absorption flux were analyzed by developing the one-dimensional mathematical modeling to the chemical absorption theory based on mass-transfer resistances in series.
Having in view the interesting and original results of this paper, which are very important for many practical applications in the modern industry. Major comments to address for consideration of publication in prestigious journal are as follows:
• How is your present study is different from
Theoretical and experimental studies of CO2 absorption by the amine solvent system in parallel-plate membrane contactors
• Abstract is too quantitative. Authors should reflect some of their results in them aiming at expanding/highlighting the contribution of their research.
• The originality/novelty is not really highlighted in Section 1 Introduction. The written paragraph focused more on what is not done, and thus will be conducted. The paragraph should elaborate more the importance, and expands/highlights the contribution of their research.
• The introduction and review of the literature is weak. It is suggested to revise the introduction by adding some of the new, latest studies from the literature and from this journal too like for example:
• An algorithm of solution should be added to the paper showing how the governing equations are solved, so that it can be clearer to the readers.
• Can the authors justify the choice of utilizing fourth-order Runge-Kutta method to obtain the numerical solution , in comparison to other traditional methods? What are the advantages over the conventional ones?
• Effects of shapes of turbulence promoters on concentration polarization is presented in table 1. But, does it depend on size of the nanoparticles. If yes/no justify.
• Research questions are needed. Note that the results in this report are typical answers to unknown questions. This is true because the manuscript provides some powerful answers to unknown questions. Note that the research questions must connect the title to the analysis of results, and conclusion. This would guide authors not to generate many results that are not consistent to provide insight. The author should update the manuscript with appropriate and relevant research questions at the end of the introduction section. This would guide the author to structure logical analysis of results. Logical questions are expected. This would help readers to link what is known in the literature with the novelty of this study.
• There are no comparisons or at least discussions with the experimental data, which does not allow assessing the quality of the results, physical correctness, and applied engineering importance.
• The correlated equation of average Sherwood number obtained numerically using the fourth Runge–Kutta method in a generalized. The method used is not explained anywhere else other than this statement. Explain how and why this method has been used.
• In Conclusion, I would suggest expanding the writing of conclusion. Instead of just highlighting the key findings, the authors should address what are their research contributions, in comparison to what are lacking for the existing literatures.
Author Response
Dear Professor Kolev,
Attached file is the revised manuscript entitled, “membranes-1638752, Theoretical and Experimental Studies of CO2 Absorption in Double-Unit Flat-Plate Membrane Contactors”, which has been submitted to Membranes (Special Issue: Membrane Separation Techniques: Advances, Challenges, and Future Avenues). It has been carefully revised and edited in which reviewers’ comments and your suggestions are incorporated. All the revisions in the revised manuscript are marked with font color in red. An Itemized Response to the Reviewers’ Comments is also enclosed.
I would like to express our thanks for the valuable comments and inputs from you and this scientific community. We hope that this revised manuscript would meet the Journal’s expectation and be considered for publication on Membranes. Your further consideration is greatly appreciated.
Best regards,
Chii-Dong Ho, Ph.D.
Vice-President for Academic Affairs
Distinguished Professor, Department of Chemical and Materials Engineering
Tamkang University
Tamsui, Taipei
Taiwan 251

Reviewer 2 Report
The paper has value to be published. My question is the design of the spacers. How are bout the resistance to flow by the introduction of spacers? will it increase the energy cost of the pump? On the other hand, some other CO2 separation methods should be introduced. in what situation, absorption method is better? why in industry application, amine absorption is most adopted but not membrane contactor?
Author Response

(The authors gave the same response as above.)

Round 2
Reviewer 1 Report
Paper can be accepted in present form